# Psychometric Evidence of the Pap Smear Test and Cervical Cancer Beliefs Scale (CPC-28) in Aymara Women from Chile

**DOI:** 10.3390/ijerph22071025

**Published:** 2025-06-27

**Authors:** Gonzalo R. Quintana, Natalia Herrera, J. Francisco Santibáñez-Palma, Javier Escudero-Pastén

**Affiliations:** 1Escuela de Psicología y Filosofía, Facultad de Ciencias Sociales, Universidad de Tarapacá, Arica 1000007, Chile; francisco.palma.santy@gmail.com (J.F.S.-P.); j.escuderopasten@gmail.com (J.E.-P.); 2Escuela de Enfermería, Facultad de Ciencias de la Salud, Universidad de Tarapacá, Arica 1000007, Chile; natalia.herrera.medina@gmail.com

**Keywords:** Aymara, indigenous, Pap test, cervical cancer, psychometrics

## Abstract

Cervical cancer (CC) remains a critical global health issue which disproportionately affects low- and middle-income countries. In Chile, the Arica and Parinacota region experiences high CC mortality and low Papanicolaou (Pap) test coverage, with indigenous Aymara women facing significant screening barriers. Understanding health beliefs surrounding CC prevention is essential for improving adherence, particularly in under-represented populations. This study assesses the psychometric properties of the CPC-28, an instrument measuring beliefs about CC and Pap testing, among Aymara women in Chile. A cross-sectional survey of 299 Aymara women (25–64) was conducted using stratified probabilistic sampling. Confirmatory factor analysis (CFA) confirmed the CPC-28’s six-factor latent structure, demonstrating strong model fit (CFI = 0.969, TLI = 0.965, RMSEA = 0.058). Reliability indices ranged from acceptable to excellent (α = 0.585–0.921; ω = 0.660–0.923). Moderate correlations emerged between severity, susceptibility, and perceived benefits of Pap testing, although CPC-28 results did not predict adherence. These findings support CPC-28’s validity evidence for Aymara women but highlight cultural influences on screening behaviors. Structural barriers, including language and healthcare inaccessibility, are likely to affect perceived susceptibility. Future research should explore indigenous perspectives and socio-cultural determinants of Pap testing, incorporating mixed-method approaches to identify culturally relevant interventions and improve screening adherence.

## 1. Introduction

World estimations determine that cervical cancer (CC) is one of the leading causes of death among women worldwide, with more than 662,000 new cases reported in 2022. Roughly 94% of the almost 350,000 deaths recorded from cervical cancer took place in low- and middle-income countries [1,2,3]. Particularly, in Chile, the CC incidence estimates about 1500 new cases per year, ranking 4th amongst all cancers [4], killing 600 women a year and primarily affecting those of reproductive and productive working age [5]. Therefore, early detection is of utmost importance for public health and research. One of the most commonly used tests for detecting precancerous lesions, regarded as the most significant achievement in CC screening, is the Papanicolaou (Pap) test [6].

A Pap smear is a screening test that collects cervical cells to check for abnormalities, infections, or precancerous changes, helping detect potential cervical cancer early [6]. It is estimated that 93% of women will undergo the PAP smear test at some point in their lifetime, as it is widely used as a screening test and prevention strategy for CC [7]. In Chile, although mortality from CC has decreased since the introduction of a prevention and control program for CC [8], this decline has been uneven across different regions of the country. Particularly, the Arica and Parinacota region not only exhibits an increasing mortality rate from CC [8] but also has one of the lowest coverage rates for Pap testing [9]. An additional factor contributing to this disparity is ethnicity, where women from Chilean indigenous groups face significant barriers to accessing preventive health services in this area. According to the latest census [10], the Arica and Parinacota region of Chile has the highest percentage of individuals belonging to indigenous groups, accounting for 35.7% of the population, with the Aymara ethnicity being the most predominant, representing over 90% of them.

In both indigenous and non-indigenous populations, adherence to the Pap test is a multifactorial phenomenon. Eggleston et al. [11] demonstrated that women who tend to fail to meet the Pap test recommendations were: (1) from racial minorities, (2) were underweight or of normal weight (vs. overweight/obese), (3) women exhibiting mood disorder symptomatology over the past two years, (4) had never been diagnosed with cancer, (5) did not meet the mammography recommendation, (6) not having had a routine check-up by a doctor in over two years, (7) women older than 35 (vs. those aged 21–35), (8) single, divorced/separated, or widowed (vs. those married or living with a partner), and (9) women with less education (vs. being a college graduate or higher). Meanwhile, racial and ethnic minority predictors of lower adherence to the PAP test comprise a lower educational level, low income, limited access to healthcare systems, poor understanding of the language used by healthcare professionals, and cultural beliefs that are predominant among indigenous populations [12,13].

In identifying risk and/or protective factors, evidence has shown that beliefs associated with CC and the Pap test play a significant role in determining who is at greater risk. Thus, the Health Belief Model (HBM) is one of the most commonly used frameworks for assessing preventive behaviors that individuals adopt in response to various health interventions [14]. The model explains the connection between an individual’s beliefs, their understanding of health threats, and the perceived barriers and benefits associated with health behaviors [15]. For instance, using a sample of 300 HIV-infected women, Lambert et al. [16] demonstrated that those with higher perceived self-efficacy and lower perceived barrier scores reported better Pap test adherence. Recent reviews on the use of the HBM applied to CC among immigrants and ethnic minority women consistently highlight that cervical cancer screening uptake is influenced by a range of factors, with cultural beliefs, knowledge, language, and access to healthcare services playing key roles [17]. In Ethiopia, a 21% screening uptake was observed, with factors such as knowledge, age, and perceived susceptibility to cancer positively impacting participation [18]. Similarly, among minority and immigrant populations in the U.S., barriers to screening included fatalistic attitudes, misconceptions about Pap smears, and limited knowledge about CC, often compounded by issues like fear of stigmatization, language differences, and difficulties accessing healthcare [19]. Meanwhile, Haitian immigrant women faced higher cervical cancer mortality rates due to low screening rates, which were driven by insufficient health education, a lack of culturally appropriate outreach, and challenges in accessing tests [20]. Altogether, these findings highlight the gap in addressing cultural misconceptions, improving knowledge about cervical cancer, and enhancing healthcare access, which could, in turn, significantly increase screening rates across diverse populations, particularly among ethnic minorities.

The HBM, while influential in public health research, demonstrates significant limitations when applied to indigenous populations such as the Aymara. Its focus on individual perceptions fails to reflect the Aymara people’s holistic view of health, which integrates spiritual, communal, and ecological dimensions [21]. The model also inadequately addresses the broader social and environmental determinants of health, including geographic isolation and inadequate infrastructure [22]. Emotional, relational, and cultural influences—which are central to indigenous health decisions—are often overlooked due to the HBM’s rationalist assumptions [23]. Applying Western-developed models without cultural adaptation risks imposing foreign values and frameworks [21], especially since standardized HBM-based instruments lack validity in these cultural settings [24]. Indigenous voices remain under-represented in HBM development, reducing its cultural relevance [25]. Ethical health interventions must consider historical injustices and systemic inequities, particularly for Aymara people [26], as well as language barriers that hinder communication and understanding [27]. The model’s individualistic orientation can also dampen community engagement, which is essential in collective cultures [28], and it generally fails to incorporate traditional health practices that are central to indigenous healing [29].

Whereas a Pap test is considered a clinical gold standard screening for CC, researchers often rely on psychometric tools to screen the presence of certain attributes. Thus, addressing the need for effective screening tools able to measure women’s health beliefs about the Pap test and CC, Urrutia and Hall [30] developed and validated the CPC-28. Based on the HBM [14], and using a sample of 333 women in the Chilean healthcare system, they reported a latent structure of six factors: barriers to taking a Pap test, cues to action, severity, the need to undergo a Pap test, susceptibility to cervical cancer, and benefits, thereby explaining 49% of the total variance and yielding adequate levels of reliability. However, despite its high value, there are no studies that evaluate their beliefs regarding CC and the associated screening test, as validated in indigenous populations. Indeed, studies addressing these issues among indigenous and ethnic populations often use samples that include immigrants to white Caucasian-dominated countries [17,18,19,20]. Indeed, much less is known about the realities of certain ethnic minority groups, particularly of indigenous and First Nations populations. Therefore, given the lack of studies on health beliefs related to CC and Pap testing among these groups—especially within the Aymara population in Chile’s Arica and Parinacota region—this study aimed to provide psychometric evidence on the validity and reliability of the CPC-28 belief instrument for assessing CC and Pap test-related beliefs among Aymara women in northern Chile.

## 2. Methods

### 2.1. Study Design

The study employed a cross-sectional design. A probabilistic sample was collected in three sequential stages. First, a stratified sample was determined from a population that included 13 indigenous associations registered with the National Corporation for Indigenous Development (CONADI) of Chile, along with a group of indigenous neighborhood associations, comprising a total of 757 female participants. Of these, 10 associations operate in urban areas, and four were in rural areas. In the second stage, considering that 56% of the women lived in urban areas and 44% in rural areas, seven urban associations were selected randomly, while all four rural associations were maintained. Finally, in the third stage, participants from both rural and urban sectors were randomly selected using the random function available in Microsoft Excel 2016 software.

It is important to note that, although probabilistic methods were used, access to the questionnaire—particularly in its digital format—may have favored participation by women with greater digital literacy, educational attainment, or health awareness. Additionally, data on the number of eligible women who declined to participate were not systematically recorded, which limits our ability to assess any potential nonresponse bias. However, several strategies were implemented to improve participation by removing common barriers (see “Procedure” below).

### 2.2. Participants

The study sample consisted of 299 Aymara women (*M*_age_ = 40.81, *SD*_age_ = 10.54) aged between 25 and 64 years. Of these, 66.55% had undergone Pap screening in the past three years, with 85.50% reporting normal results in their most recent test. Occupational status indicated that 53.79% were active in the labor force, with 27.24% being unemployed. The majority of the sample (52.92%) were single, and 60.34% were in a relationship. Educational attainment varied, with secondary school completion predominating (48.29%). Income levels primarily ranged between CLP 250,000 and 500,000, representing 52.67% of the participants. Primary healthcare was provided through the national health program (FONASA; 91%). These demographic data indicate a balanced representation across several key variables, reflecting the profile of the target community (see Table 1).

### 2.3. Sociocultural Context

Overall, Aymara women in Chile’s Arica y Parinacota region often face challenges in terms of accessing healthcare services due to geographic isolation, language barriers, and cultural practices. While public health facilities like the Hospital Regional Dr. Juan Noé and various rural clinics provide services, their reach is limited in remote areas, affecting timely access to preventive care such as Pap smears. Language differences, particularly among older women who primarily speak Aymara, can hinder effective communication with healthcare providers, leading to misunderstandings and reduced trust in medical services [31]. Still, Chile provides a translator and cultural moderator for its aboriginal people through the Special Program for Health and Indigenous Peoples (PESPI) (https://ssms.gob.cl/como-me-cuido/programas-de-salud/programa-especial-de-salud-y-pueblos-indigenas-pespi/) (accessed on 8 April 2025).

Culturally, health is perceived holistically, intertwining physical, spiritual, and social well-being. Traditional healers, known as yatiri, play a significant role in health practices, often using rituals and herbal remedies [32]. This worldview may influence perceptions of Western medical procedures, sometimes leading to skepticism or alternative health-seeking behaviors. Organizations like the Asociación Nacional de Mujeres Rurales e Indígenas (ANAMURI) work to empower Aymara women, advocating for culturally sensitive healthcare approaches that respect and integrate traditional beliefs [26].

## 3. Measures

### 3.1. Adherence to Pap Smear Screening

Adherence to cervical cancer screening was assessed using four questions. The first was: “Have you had a Pap smear test in the last three years?”—a definition used by the Chilean public health system to determine Pap smear test coverage [4]—with dichotomous response options (i.e., Yes = 1, No = 2). Participants were then asked about the results of their most recent Pap smear test, with response options including “Normal”, “Abnormal”, or “Indeterminate” [4]. Those who reported an abnormal result were asked to specify the outcome of their most recent test (e.g., “Inflammatory lesion and infection”). Finally, participants were asked about the frequency of their Pap smear tests, with response options including: “Annually”, “Every 2 years”, “Every 3 years” [4], and “Whenever I remember.”

### 3.2. Beliefs About Pap Smear and Cervical Cancer (CPC-28)

The Beliefs about Pap Smear and Cervical Cancer (CPC-28) test, developed by Urrutia and Hall [30], is a questionnaire based on the HBM. This instrument consists of 28 items, distributed across six dimensions: barriers to Pap smear testing (nine items; e.g., “I do not get a Pap smear because I have to wait a long time to be seen”), cues to action (six items; e.g., “because family members told me to get it”), benefits of the test (three items; e.g., “to take care of my health”), the perceived need for the test (three items; e.g., “if I have no symptoms or discomfort, I do not need a Pap smear”), perceived severity of cervical cancer (four items; e.g., “cervical cancer is a serious problem”), and perceived susceptibility to cervical cancer (three items; e.g., “I am at risk of developing cervical cancer”; see Appendix A). The scale uses a 4-point Likert scale ranging from strongly agree (1) to strongly disagree (4). The reliability of the CPC-28, measured by Cronbach’s alpha, has previously shown a value of 0.74, indicating adequate internal consistency for assessing beliefs related to cervical cancer and Pap smear testing [30].

## 4. Procedure

To address the challenge of administering the questionnaire to women, the CPC-28 was converted into a digital format. This allowed the questionnaire to be distributed via a link sent to the phone contacts of selected women who preferred the digital version. For those who requested a paper version, it was delivered to their homes by the president or a member of their organization. A three-week deadline was set for completing the digital questionnaire. Data collection, both digital and paper-based, took place between April and June 2021. Prior to distributing the questionnaires in late March 2021, a Zoom training session was conducted for three women from different Aymara associations in the region. The purpose of this training was to familiarize them with the questionnaire so they could assist others in completing it. Once the paper questionnaires were completed, one of the trained women from the associations collected them from the participants’ homes and delivered them to the principal investigator. The estimated time to complete the paper questionnaire was approximately 45 min, while the digital version took around 30 min.

The inclusion criteria were twofold: being between 25 and 64 years old and belonging to one of the 14 indigenous associations recognized by CONADI. Women diagnosed with cervical cancer and/or those who had undergone a hysterectomy were excluded from the study. 

Informed consent was obtained from all women prior to their inclusion in the study. Additionally, an intercultural facilitator was provided upon request to support participants in understanding the requirements and to apply the principle of autonomy in terms of their decision to participate.

## 5. Statistical Analysis

The statistical plan comprised the following steps: data imputation, descriptive analysis, dimensionality testing, predictive validity testing, and reliability testing, in compliance with EQUATOR’s recommendations for reporting the results of studies including instrument and scale development and testing [33].

### 5.1. Missing Data

We used the Full Information Maximum Likelihood (FIML) method to handle missing data. FIML estimates model parameters using all available information from observed variables, accommodating missing values through an iterative procedure. This method allows for the inclusion of all participants in the analysis, regardless of missing data, and provides unbiased estimates under the assumption that the missing data mechanism is random [34]. Missing values in the CPC-28 items ranged from 2.3% to 10.7%, while Little’s test [35] demonstrated that the missing data were completely random [MCAR; χ^2^ = 1140.329, df = 1145, *p* = 0.53].

### 5.2. Descriptive Analysis

First, we used central tendency and dispersion statistics to describe the sample, as well as the items and dimensions of the CPC-28 test (i.e., *M*, *SD*, *n*, and %; see Table 1 and Table 2, respectively).

### 5.3. Construct Validity

To determine the construct validity of the CPC-28 model, we conducted a confirmatory factor analysis (CFA) using the Weighted Least Squares Mean and Variance adjusted (WLSMV) estimation method, as recommended [36,37,38].

Model fit was assessed using the Comparative Fit Index (CFI), the Tucker-Lewis Index (TLI), the Root Mean Square Error of Approximation (RMSEA) with its 90% confidence interval, and the Standardized Root Mean Square Residual (SRMR). CFI and TLI values above 0.90 are considered acceptable, and values above 0.95 are considered good, while RMSEA and SRMR values of 0.05 and 0.08 or lower indicate a good and an acceptable fit, respectively [39,40]. Factor loadings were evaluated based on the criteria suggested by Jöreskog and Sörbom [41], where loadings greater than 0.7 are considered optimal, and those between 0.7 and 0.3 are considered adequate. Additionally, the strength of correlations between dimensions was assessed using Cohen [42] criterion, classifying correlations above 0.7 as strong and those between 0.7 and 0.3 as moderate.

### 5.4. Reliability Evidence

Reliability of each dimension and the overall scale was assessed using Cronbach’s alpha and McDonald’s hierarchical omega for a more effective reliability evaluation [43,44,45]. In general, a reliability level of ≥0.7 is considered satisfactory. However, it should be noted that this threshold serves more as a guideline than as a strict empirical rule [46].

### 5.5. Predictive Validity

To examine whether cervical cancer-related health beliefs predicted the likelihood of undergoing a Pap smear test, we conducted a binary logistic regression analysis using a forced-entry method. The dependent variable was Pap test uptake (“*Have you undergone a Pap smear in the last three years?*”). The independent variables included the five dimensions of the CPC-28: perceived barriers, cues to action, perceived benefits, perceived severity, and perceived susceptibility. In addition, we included participants’ age and education level as covariates. Prior to running the regression analysis, assumption checks were conducted to evaluate multicollinearity using Variance Inflation Factors (VIFs), and the linearity of the logit for continuous variables was assessed using the Box-Tidwell procedure. All VIF values were below the conventional threshold of 5, indicating no problematic multicollinearity: Barriers (VIF = 1.33), Cues (VIF = 1.23), Benefits (VIF = 1.52), Severity (VIF = 1.33), Susceptibility (VIF = 1.29), age (VIF = 1.15), and education level (VIF = 1.17). Moreover, the assumption of linearity between continuous predictors and the logit of the dependent variable was evaluated using the Box-Tidwell procedure [47]. None of the interaction terms between predictors and their log-transformed counterparts were statistically significant (*p*s > 0.15), suggesting that the linearity assumption of the logit was met. Therefore, our findings confirm that the assumptions for conducting logistic regression were satisfied.

A binary logistic regression was conducted to examine whether cervical cancer-related beliefs (Barriers, Cues, Benefits, Severity, and Susceptibility), age, and education level predicted the likelihood of Aymara women undergoing a Pap smear test, to evaluate the strength and direction of associations. Model fit was assessed via the likelihood ratio chi-square (*χ*^2^) test, pseudo *R*^2^ values, and classification accuracy. In addition, we compared mean changes in CPC-28 items and dimensions using *t*-tests and Cohen’s *d* effect size, wherein 0.10 was considered small, 0.30 medium, and 0.50 or greater large [42] (see Appendix A). Logistic regression analyses were conducted using Python (v. 3.13.5), whereas the rest were conducted using Mplus (version 8.11) [48].

## 6. Results

Table 3 presents the CFA analysis results of the model fit. The evaluated model exhibited a good fit to the data, WLSMV *χ*^2^ (125) = 667.413; *p* < 0.001; CFI = 0.969; TLI = 0.965; RMSEA = 0.058; 90% CI = 0.052–0.065.

Table 3 displays the standardized factor loadings and reliability indices for the six-factor CFA model. The standardized factor loadings ranged from 0.961 to 0.355 (*p*s < 0.001). The reliability indices for alpha (*α*) ranged from 0.585 to 0.921, while for omega (*ω*), they ranged from 0.660 to 0.923.

Regarding the significant correlations between factors, these ranged from weak (*Φ* = −0.288) to strong (*Φ* = 0.777). The barrier dimension was negatively correlated with the benefits (*Φ* = −0.288; *p*s < 0.001) and positively correlated with the need to undergo a Pap smear (*Φ* = 0.670; *p*s < 0.001). Additionally, the action cues dimension was positively correlated with the benefits (*Φ* = 0.320; *p*s < 0.001) and the severity of cervical cancer (*Φ* = 0.306; *p*s < 0.001). Furthermore, the benefits dimension was negatively correlated with the need to undergo a test (*Φ* = −0.454; *p* < 0.001) and positively correlated with the severity (*Φ* = 0.727; *p* < 0.001) and susceptibility (*Φ* = 0.624; *p* < 0.001) of cervical cancer. The dimension of the need to undergo a Pap smear was negatively correlated with the severity (*Φ* = −0.390; *p* < 0.001) and susceptibility (*Φ* = −0.219; *p* < 0.001) of cervical cancer. Finally, the severity dimension was positively correlated with the susceptibility (*Φ* = 0.777; *p* < 0.001) of cervical cancer.

Table 4 presents the Spearman’s Rho correlations among the CPC-28 dimensions. Correlations varied between null-to-moderate across the CPC-28 dimensions. Among them, “Need to take a Pap test” correlated significantly and positively with “Barriers to taking a Pap test”, “Severity of cervical cancer” correlated significantly and positively with “Benefit of taking a Pap test”, and “Susceptibility to cervical cancer” correlated significantly and positively with “Severity of cervical cancer”, with all of them showing a moderate magnitude.

Finally, a binary logistic regression analysis was conducted to examine whether cervical cancer-related beliefs, age, and education level predicted the likelihood of Aymara women undergoing a Pap smear test. The overall model was not statistically significant, *χ*^2^ (7) = 5.62, *p* = 0.682, indicating that the predictors did not significantly improve the prediction of Pap test uptake compared to a null model. The model explained only a small proportion of variance (Nagelkerke *R*^2^ = 0.018) and correctly classified a limited number of cases. None of the individual predictors reached statistical significance (*p*s > 0.13), although barriers to screening approached marginal significance (*B* = 0.04, SE = 0.03, *p* = 0.14). This suggests a possible trend in which higher perceived barriers may be associated with the increased likelihood of not getting tested, but the evidence is not strong enough to draw firm conclusions.

## 7. Discussion

Cervical cancer (CC) remains a major global health concern, particularly in low- and middle-income countries, where the majority of deaths from CC occur. In Chile, CC ranks among the leading causes of cancer-related deaths, with significant regional disparities in screening coverage [8], particularly in the Arica and Parinacota region, home to a high percentage of Aymara indigenous women [10]. Adherence to the Pap smear test, the primary screening tool for CC, is influenced by multiple factors, including socioeconomic status, healthcare access, education, and cultural beliefs, with indigenous and minority populations facing additional barriers such as language difficulties and healthcare mistrust [49]. The Health Belief Model (HBM) has been widely used to understand screening behaviors, demonstrating that perceptions of susceptibility, self-efficacy, and barriers strongly impact treatment adherence. Despite existing research on minority and immigrant populations, little is known about the specific health beliefs of indigenous groups, particularly in non-Western contexts. The CPC-28, a psychometric tool developed to assess beliefs about CC and Pap testing, has been validated in the Chilean healthcare system but has not yet been tested among indigenous populations. Therefore, this study sought to provide psychometric evidence as to the validity and reliability of the CPC-28 among Aymara women in northern Chile in the hope of contributing to a more culturally informed approach to CC prevention, while addressing ethnic inequalities in test coverage and the use of women’s cancer screening [50].

The CPC-28 instrument demonstrated good construct validity based on confirmatory factor analysis (CFA), where all goodness-of-fit statistics indicated a good to excellent fit, meaning that the model closely represents the data. This demonstrates that the six-factor solution found by Urrutia and Hall [30] aligns well with the Aymara people in the north of Chile. Unlike Urrutia and Hall [30], our study provides much stronger evidence of validity for the CPC-28 since they not only used an exploratory factor analysis (EFA) along with the Kaiser–Meyer–Olkin (KMO) test, without any goodness-of-fit estimator. On the one hand, an EFA is a preliminary step for a theory-based CFA, Urrutia and Hall [30] only conducted an EFA, without testing their own latent solution, subsequently. EFA is a data-driven technique used when the underlying factor structure of a questionnaire is unknown, whereas a CFA is a theory-driven approach that tests a hypothesized factor structure [38]. Moreover, EFA relies solely on factor loadings and eigenvalues, whereas the CFA and goodness-of-fit estimators indicate how well the proposed factor structure represents the data. Finally, a KMO test assesses sampling adequacy for factor analysis but does not confirm whether the extracted factors truly represent the intended constructs [51]. Interestingly, Guvenc et al. [52] also evaluated the HBM models when applied to CC and the Pap smear test in Turkish women, offering a different psychometric scale. However, they also tested their scale using an EFA, KMO, and Bartlett’s test. Therefore, our findings present strong psychometric evidence of construct validity for the CPC-28, corroborating Urrutia and Hall [30]’s six-factor structure.

The six-factor solution for the CPC-28 offers supporting evidence for Rosenstock’s HBM. These dimensions included perceived barriers to undergoing the Pap test, cues to action for taking the test, the perceived benefits of the Pap test, the necessity of taking the Pap test, the perceived severity of cervical cancer, and susceptibility to developing cervical cancer [53]. As previously mentioned, the HBM seeks to explain people’s individual behavior in response to preventive health interventions by examining their underlying beliefs [18,54]. Studies conducted among indigenous and non-indigenous women suggest that these dimensions not only represent women’s beliefs about preventive health actions, but can also predict their behavior regarding Pap test adherence [18,30]. For instance, Filipino women who perceived themselves as being susceptible to CC or who considered the disease to be severe were 3.81 and 2.6 times more likely, respectively, to undergo Pap testing [50]. This exemplifies that perceptions of susceptibility and severity significantly influence the likelihood of undergoing Pap screening. Therefore, these findings demonstrate that the CPC-28 six-factor solution is able to capture the HBM in both Aymara and non-indigenous amongst Chilean women, enabling future research from the theoretical perspective of the HMB. Future studies may evaluate how well this solution holds up when tested in a similar ethnic group from different parts of the Andina macro-region, such as the Aymara people from Perú or Bolivia, or different ethnic groups from Chile, such as the Mapuche people. It is likely that socio-economical differences would modulate the experience of access to sexual health services [50,55,56], likely depending also on the official recognition and special protections these groups may have depending on their countries of origin and the accessibility the health system has in place (for a list of the recognized Chilean first nation communities, see https://www.chileatiende.gob.cl/instituciones/AI002; accessed on 8 April 2025).

Analysis of the standardized factor loadings in the Aymara women population revealed that all 28 items were organized into the six factors or dimensions outlined in the questionnaire. Items exhibited optimal loadings, reinforcing the notion that beliefs about the Pap test and cervical cancer are multifactorial phenomena in a similar way between indigenous and non-indigenous women [50,53]. Meanwhile, reliability indices indicated acceptable-to-excellent reliability levels, higher than those found by Urrutia and Hall [30], with no substantial differences between statistical indices.

On the one hand, the dimension “severity” reached the highest reliability and factor loading. This is consistent with previous findings since cancer is generally perceived as a severe and serious disease [50]. On the other hand, “susceptibility” exhibited the lowest reliability scores, meaning that these items are not consistently capturing the same underlying concept, leading to inconsistent responses, reduced measurement precision, weaker statistical associations, or limited interpretability. Particularly, item 26—“*I am at risk of developing cervical cancer*”—demonstrated a moderate factor loading and reliability. While removing this item might improve the instrument’s reliability quantitatively, a qualitative analysis is warranted to consider its cultural relevance, especially if the scale is to be tested with other ethnic groups. Therefore, we chose to keep it as part of the scale. We decided to leave the scale as it is for two main reasons: (1) we seek future studies to corroborate these results with a different sample of Aymara women, while still strongly encouraging researchers to remove item 26 should it replicate the same trend; (2) allowing other studies researching Pap test adherence in different Chilean aboriginal groups to evaluate the fitness of item 26 in those samples. Furthermore, these findings may reflect the Aymara women’s low perceived susceptibility to CC, which could be explained by lower education levels and lower access to the health services that would supply this information, which consequently results in lower awareness among their own community, perpetuating a vicious cycle of structural differences in access to appropriate sexual health support and education [11,12,13]. For instance, a study conducted among Aymara, Quechua, and Afro-descendant women in Peru indicated that those with low or incorrect knowledge about cervical cancer causation perceived themselves as being less susceptible to the disease [50]. A similar pattern was observed in a study of rural Chinese women, wherein those with limited knowledge exhibited lower Pap test adherence due to their low perceived susceptibility to cervical cancer [57,58]. It is known that Aymara and other indigenous women have reduced healthcare access in comparison to non-indigenous women [4], which diminishes their opportunities to receive health education from professionals.

A moderate positive correlation was observed between the “perceived severity of cervical cancer” and “perceived benefits of the Pap test”, as well as “perceived susceptibility to cervical cancer”. This suggests that as women’s awareness of the severity of cervical cancer increases, so does their recognition of the benefits of Pap testing in terms of early detection. This is consistent with the findings of a study among the indigenous American Indian Zuni women, where greater knowledge regarding cervical cancer transmission was associated with higher Pap test adherence [59]. Moreover, a moderate positive correlation was found between the “perceived need for Pap testing” and “perceived barriers to undergoing the procedure”. This relationship may be explained by the fact that Aymara women reside in rural locations, language barriers due to that most health care providers do not speak the same language and translators are highly scarce, leading to lower adherence to Pap screening and perpetuating structural issues by Aymara women receiving limited preventive education, meaning that they fail to understand the provided healthcare information [60,61]. Furthermore, a moderate correlation was found between “severity of cervical cancer” and “susceptibility to cervical cancer”, suggesting that Aymara women who recognize CC as a serious disease are more likely to see themselves at risk. Structural barriers such as language difficulties and rural healthcare inaccessibility contribute to low awareness and preventive action [4]. Given the strong role of community in Aymara health beliefs, increasing awareness of cervical cancer’s severity may help improve risk perception, but interventions must address cultural and structural barriers to be effective [62]. A similar influence of social support, whether facilitating or hindering preventive action, was also noted in a study of indigenous women in Ecuador, where family opposition to Pap testing was perceived as a barrier [63].

In terms of predictive validity, the CPC-28 did not distinguish between Pap test adherence and the perceptions of Aymara women regarding perceived barriers, cues to action, perceived benefits, the necessity of Pap testing, perceived severity, or susceptibility to cervical cancer. Consequently, the CPC-28 is not able to predict Pap test adherence in Aymara women. This finding contrasts with studies conducted among indigenous women in China, the Philippines, and Peru, where higher perceived barriers were associated with lower Pap adherence, while higher perceived benefits and greater perceived severity of cervical cancer increased Pap testing likelihood [50,57,58]. This discrepancy may stem from cultural beliefs among Aymara women that are not captured by the HBM, which ultimately determine their Pap adherence behaviors.

The CPC-28 was devised with urban, Spanish-speaking Chilean women in mind [30]. Its direct deployment in an indigenous setting—even with the assistance of intercultural facilitators—fell short of current cross-cultural adaptation standards, which emphasize iterative forward–backward translation, cognitive debriefing, and differential item functioning analysis to secure semantic, conceptual, and metric equivalence [64,65]. Constructs such as “individual susceptibility” or “self-efficacy” derive from a biomedical, individualist logic that may not map onto the Aymara people’s relational, holistic model of health [21,23]. The marginal reliability of the susceptibility subscale and the poor loading of item 26 could plausibly reflect this cultural incongruence. Future work should, therefore, undertake a full, participatory adaptation—ideally incorporating Aymara terminology and emic dimensions such as ayni (i.e., reciprocity) or spiritual causation—to enhance content validity and minimize construct bias [24].

Although a stratified, multi-stage probability design was employed, the sampling frame was limited to women affiliated with registered indigenous associations and reachable through digital or home-visit modalities. Census data indicate that roughly 40% of Aymara women in Arica-Parinacota are unaffiliated and many reside in high-altitude hamlets with limited connectivity [10]. Coupled with the absence of refusal tracking, this restriction likely produced a sample skewed toward socially organized, digitally literate, and health-engaged individuals—evidenced by Pap-test coverage (66%) that exceeds the regional average of 52% [8]. Consequently, the factorial structure and reliability coefficients reported here should not be uncritically generalized to non-affiliated, highly rural, or trans-border Aymara populations, who often experience stronger linguistic, geographic, and structural barriers [66]. Replication studies using household-based or respondent-driven designs—especially in Peruvian and Bolivian Aymara communities—are needed before advocating the CPC-28 as a pan-Andean metric.

The strength of this study relies on its design, sample, and psychometric scale and analyses used to assess its validity. However, this study was not without limitations. The use of a cross-sectional design precludes causal inferences between Pap adherence and Aymara women’s beliefs, much like the lack of a qualitative mix design would have allowed us to explore Aymara women’s beliefs from their own cultural perspective, particularly those related to Pap test adherence. More importantly, whereas we used a well-known predictive criterion, the study could have included others [67]. Future research employing longitudinal or mixed-methods approaches could provide deeper insights into how these beliefs evolve over time and influence adherence behaviors. Additionally, the absence of qualitative methods limits our ability to explore Aymara women’s beliefs from their own cultural perspectives—particularly those related to perceptions of cervical cancer and Pap test adherence. Incorporating qualitative data would be especially useful in contextualizing culturally specific constructs and identifying nuances not captured by standardized scales.

Due to the community-based and multimodal nature of data collection, and in line with respecting participant autonomy in culturally sensitive contexts, information on the number of eligible women who declined participation or did not complete the questionnaire was not systematically collected. This limits our ability to generate a recruitment flowchart or quantify refusal rates, which we acknowledge as a limitation in assessing potential selection bias. Moreover, given the face-to-face recruitment conducted by community leaders, responses may have been subject to social desirability bias [68]. Future anonymous, self-administered formats are recommended.

A further limitation is the relatively low internal consistency of the “susceptibility” subscale (α = 0.585), which suggests that the items may not adequately capture the intended construct for this population. Item-level inconsistencies indicate the need for subscale refinement, ideally informed by qualitative research to culturally adapt the concept of “susceptibility” in ways that are meaningful for Aymara women.

While we employed a widely used predictive criterion, Pap smear adherence, the CPC-28 dimensions did not significantly predict this outcome. This suggests that the scale, in its current form, may not fully account for the range of factors influencing adherence in this population. Future studies should consider including additional variables such as health literacy, social influence, and logistical access to care, which may enhance the model’s predictive utility. Moreover, research with more diverse indigenous groups is warranted, particularly when examining how cultural frameworks shape health beliefs and behaviors. Including a broader set of criterion variables will be crucial for advancing the validity and applicability of culturally sensitive health belief models.

## Figures and Tables

**Table 1 ijerph-22-01025-t001:** Sociodemographic statistics of the sample.

*N* = 299	*M*	*SD*		*n*	%
**Age**	40.81	10.54	**Relationship status**		
			In a relationship	178	60.34
**Occupational category**	*n*	%	Single	117	39.66
Inactive	55	18.97	**Average monthly income (CLP)**		
Active	156	53.79	<250,000	53	20.23
Unemployed	79	27.24	250,000–500,000	138	52.67
**Marital status**			500,000–1,000,000	50	19.08
Single	154	52.92	1,000,000–3,000,000	21	8.02
Common law	5	1.72			
Married and/or engaged	84	28.87	**PAP-related indicators**		
De facto separated	23	7.90	*Have you undergone a Pap smear in the last three years?*		
Divorced	23	7.90	Yes	199	33.44
Widowed	2	0.69	No	100	66.55
**Level of education**			*How often do you take a Pap smear?*		
Early childhood education	2	0.68	Annually	59	21.77
Primary education	31	10.62	Every 2 years	66	24.35
High school	141	48.29	Every 3 years	62	22.88
Advanced technical studies	72	24.66	*Result of the most recent Pap smear*		
University professional	46	15.75	Normal	224	85.50
**Health care provider**			Altered	14	5.34
Public (Fonasa)	263	91.00	Did not know the result	24	9.16
Private (Isapre)	19	6.57	*Abnormality was identified in the most recent Pap smear*		
Dipreca	1	0.35	Inflammatory lesion	5	33.30
None	6	2.08	Infection	10	66.70

*N* = Total sample size; *n* = group size; % = Percentage; *M* = Mean; *SD* = Standard deviation; Fonasa = National Health Fund; Isapre = Private Health Prevision Institutions; Dipreca = Chilean Police’ Retirement Funds Administration Office.

**Table 2 ijerph-22-01025-t002:** CPC-28 items and dimensions reliability and factor loadings.

Items	Descriptive Statistics			Factor Loadings						Reliability	
	*M* (*SD*)	Skew	Kurt	BA	CA	BP	NP	SC	SU	α †	ω †
Item 1	2.66 (0.91)	−0.25	−0.68	0.681						0.873	0.873
Item 2	3.10 (0.80)	−0.88	0.60	0.599						0.879	0.880
Item 3	3.09 (0.83)	−0.78	0.27	0.776						0.869	0.870
Item 4	2.88 (0.94)	−0.57	−0.47	0.738						0.869	0.870
Item 5	2.68 (0.98)	−0.37	−0.83	0.810						0.866	0.866
Item 6	2.59 (1.07)	−0.18	−1.21	0.869						0.856	0.857
Item 7	2.66 (1.02)	−0.26	1.03	0.815						0.860	0.861
Item 8	2.90 (0.90)	−0.52	−0.53	0.698						0.878	0.879
Item 9	2.72 (1.02)	−0.28	−0.98	0.735						0.871	0.872
Item 10	2.05 (0.89)	0.44	−0.61		0.714					0.850	0.855
Item 11	2.05 (0.85)	0.50	−0.33		0.723					0.839	0.847
Item 12	2.20 (0.91)	0.34	−0.68		0.739					0.830	0.832
Item 13	2.11 (0.89)	0.36	−0.76		0.748					0.824	0.828
Item 14	2.22 (0.94)	0.29	−0.81		0.863					0.810	0.812
Item 15	2.24 (0.94)	0.22	−0.89		0.879					0.802	0.804
Item 16	1.39 (0.51)	0.63	−1.06			0.748				0.551	0.552
Item 17	1.25 (0.46)	1.65	3.17			0.884				0.614	0.615
Item 18	1.22 (0.49)	2.32	5.80			0.803				0.616	0.618
Item 19	3.13 (0.78)	−0.92	0.94				0.897			0.663	0.665
Item 20	3.07 (0.84)	−0.80	0.27				0.784			0.709	0.709
Item 21	3.10 (0.77)	−0.80	0.63				0.806			0.785	0.786
Item 22	1.30 (0.56)	2.04	4.86					0.961		0.882	0.884
Item 23	1.34 (0.62)	1.91	3.94					0.915		0.915	0.916
Item 24	1.37 (0.62)	1.75	3.14					0.894		0.900	0.904
Item 25	1.34 (0.62)	2.07	4.96					0.945		0.891	0.894
Item 26	2.22 (1.03)	0.32	−1.06						0.355	0.725	0.725
Item 27	1.62 (0.73)	0.99	0.57						0.851	0.384	0.403
Item 28	1.61 (0.73)	1.05	0.64						0.821	0.382	0.401
		*M*	*SD*	**Correlations**						α index	ω index
Barriers—BA		25.28	6.10	–						0.882	0.883
Cues to action—CA		12.86	4.10	0.040	–					0.851	0.854
Benefits—BP		3.86	1.14	−0.288 *	0.320 *	–				0.688	0.691
Need—NP		9.29	2.01	0.670 *	0.078	−0.454 *	–			0.795	0.800
Severity—SC		5.35	2.15	−0.095	0.306 *	0.727 *	−0.390 *	–		0.921	0.923
Susceptibility—SU		5.45	1.86	−0.008	0.126	0.624 *	−0.219 *	0.777 *	–	0.585	0.660

BA = Barriers to take a Pap; CA = Cues to action to take a pap test; BP = Benefit to take Pap test; NP = Need to take a Pap test; SC = Severity of cervical cancer; SU = Susceptibility to cervical cancer; *M* = Mean; *SD* = Standard deviation; Skwe = Skewness; Kurt = Kurtosis; * = *p* < 0.001. † = if the item is dropped.

**Table 3 ijerph-22-01025-t003:** Summary of fit indices for confirmatory factor solutions of the CPC-28.

Model	Par	*χ* ^2^	df	*χ*^2^/df	RMSEA	90% CI	CFI	TLI	SRMR
CFA—6 factors	125	667.413 *	335	1.992	0.058	0.052–0.065	0.969	0.965	0.063

Par = Free Parameters; χ^2^ = Chi-square; df = Degrees of Freedom; RMSEA = Root Mean Square Error of Approximation; CI = 90% Confidence Interval; CFI = Comparative Fit Index; TLI = Tucker-Lewis Index; SRMR = Standardized Root Mean Square Residual; * = *p* < 0.001.

**Table 4 ijerph-22-01025-t004:** Pap smear test adherence over the last three years comparison by CPC-28 items.

	BA	CA	BP	NP	SC	SU
Barriers	−					
Cues	−0.002	−				
Benefit	−0.187 **	0.246 ***	−			
Need	0.499 ***	0.03	−0.332 ***	−		
Severity	−0.006	0.213 ***	0.483 ***	−0.291 ***	−	
Susceptibility	0.008	0.039	0.352 ***	−0.089	0.492 ***	−

Note: Spearman’s Rho; ** *p* < 0.01, *** *p* < 0.001. BA = Barriers to taking a Pap test; CA = cues to action to take a pap test; BP = benefit of taking a Pap test; NP = need to take a Pap test; SC = severity of cervical cancer; SU = susceptibility to cervical cancer.

## Data Availability

The data is publicly available at https://osf.io/sy8p4/files/osfstorage (accessed on 8 April 2025).

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
