# Peer review of "Psychometric Evidence of the Pap Smear Test and Cervical Cancer Beliefs Scale (CPC-28) in Aymara Women from Chile"

_ijerph, 2025, doi:10.3390/ijerph22071025_

Round 1
Reviewer 1 Report
Comments and Suggestions for Authors
My Comments:
This manuscript titled "Psychometric Evidence of the Pap Smear Test and Cervical Cancer Beliefs Scale (CPC-28) in Aymara Women Sample from Chile" addresses cervical cancer screening adherence among Aymara women in Chile that is a vital public health issue using a validated psychometric tool based on the Health Belief Model (HBM). The study is timely, relevant, and grounded in solid methodology. The study is valuable that contributes meaningfully to literature on cultural disparities in preventive health behavior. However, there are some limitations in the study, and several recommendations are provided that should be addressed before publication in the International Journal of Environmental Research and Public Health. These limitations are outlined below:
Limitations & Suggestions
- Predictive Validity Limitation: The CPC-28 dimensions did not significantly predict Pap smear adherence. Future studies should consider incorporating additional variables such as health literacy, social influence, and logistical access to enhance the predictive capacity of the model.
- Low Reliability of the Susceptibility Subscale: The Cronbach’s alpha for the susceptibility dimension (α = 0.585) indicates low internal consistency, with notable item-level inconsistencies. A refinement of this subscale is recommended, potentially supported by qualitative research to culturally adapt the concept of "susceptibility" for Aymara women.
- Cross-Sectional Design Constraints: The current cross-sectional approach limits causal interpretations. A longitudinal or mixed-methods design in future research could provide more comprehensive insights.
- Need for Deeper Cultural Context: While cultural barriers are acknowledged, the study would benefit from richer ethnographic detail such as narratives exploring Aymara women’s perceptions of health, the body, and gender roles.
- Writing Clarity: Certain sections, particularly the Introduction and Discussion, would benefit from more concise and focused writing to improve readability and flow.
- Table Formatting: Ensure consistent formatting and clear, descriptive titles across all tables to enhance clarity and presentation quality.
- Supplementary Material Integration: Key elements of the supplementary material such as representative item examples from the CPC-28 should be summarized or referenced within the main text to aid reader comprehension.
Author Response
We appreciate the reviewer’s comments and feedback which are insightful and definitely improves our study. We included the reviewer’s suggestions #1 to #3 in our limitation section, as well as point #6 and #7. Furthermore, while we strongly agree with the reviewer’s point #4, we believe the study scope is too specific (i.e., psychometric) to delve deeper in the points the reviewer mentions. The study focus is the scale psychometric properties and evidence of validity, and in incorporating the other reviewer’s comments, the manuscript length is exciding one where the reader may become distracted from too much information being addressed. Finally, our introduction has been significantly modified following the reviewer’s point #5 along with other reviewer’s comments.
The authors kindly invite the reviewer to also read the other reviewer's comments, and our answers to them, for many of your suggestions were also addressed, and sometimes more directly, in the other reviewer comments responses.
Reviewer 2 Report
Comments and Suggestions for Authors
Attached in file

Author Response
The authors sincerely appreciate the detailed and insightful comments from the reviewer. These were of significant help to improve our study. We answer to each point main below following the structure given by the reviewer.
Introduction
First, we believe the mentioning of the results is to the “Plain-language Summary” required by the journal. We will correct that with the Journal. Furthermore, we incorporated a new paragraph discussing the limitation of the HBM when using it in an indigenous group, particularly the Aymara people.
Methods
The reviewer is right regarding the broader context. We provided a sociocultural context related to the sample and their characteristics to broaden the reader’s understanding as to how these factors may limit our sample recruitment and the scope of the results.
We thank the reviewer for this important observation. We fully agree that the potential for participation bias—particularly favoring more educated or health-aware women—is a valid concern. While we employed probabilistic sampling and multiple modes of data collection (digital and paper-based) to reduce barriers to participation, we acknowledge that the digital format may have introduced accessibility-related bias. Additionally, we regret that we were unable to systematically track the number of women who declined to participate, which limits adherence to certain aspects of the STROBE guidelines. In response, we have modified the manuscript to explicitly acknowledge these limitations in the sampling method section. We believe that this transparent statement provides important context for interpreting our findings and strengthens the overall methodological rigor of the paper, even in the absence of this data.
Results
Regarding the recruitment process, we thank the reviewer for their thoughtful comment. We acknowledge that the lack of a participant flowchart and detailed reporting of eligibility and refusal rates limits full transparency in the recruitment process, as outlined in the STROBE guidelines. Unfortunately, due to the decentralized nature of our recruitment—conducted through indigenous associations with both digital and paper-based modalities—and the sociocultural context in which the study was implemented, it was not feasible to systematically track the number of women who declined participation or were lost to follow-up.
However, we indeed made efforts in the revised manuscript to clearly describe our multi-stage probabilistic sampling procedure and the logistics of participant recruitment. Specifically, the manuscript now includes an explicit acknowledgment of the potential for nonresponse bias and clarifies that women may have been more likely to participate if they were more digitally literate or health-conscious. Additionally, we highlight the culturally tailored strategies implemented to reduce structural barriers to participation—such as in-home paper survey delivery by community leaders, bilingual support through intercultural facilitators, and training of local women to assist participants. Furthermore, while we agree that including a participant flow diagram would enhance transparency, we respectfully note that our study was designed primarily for psychometric validation, not clinical follow-up or intervention, and thus had no longitudinal components or attrition. We believe that the described recruitment procedures and the representative sampling strategy help mitigate selection bias and ensure the generalizability of our findings within the Aymara population in this region.
Still, we also acknowledge this in the limitation section.
Regarding conducting logistic regression, we agreed with the suggestion, and now included a logistic regression analysis testing age, education level, and the CPC-28 scores associated with screening behaviour.
Discussion
We structured a discussion section that first restates the study topic, problematization, and goal, as well as its main results, to later on go into discussing each of them. We believe this helps the reader to remember the purpose of the study and comprehend the results within the scope of the literature. Thus, under this structure, we reserve discussing the limitations at the end.
We have revised the discussion section to incorporate a more critical reflection on the study’s methodological and conceptual limitations. In this updated version, we address concerns related to predictive validity, cross-cultural applicability of the CPC-28, issues of generalizability, and potential sources of bias. While these points are treated in a summarized fashion to maintain the flow of the discussion, we have taken care to ensure that the limitations are acknowledged and that the conclusions are now presented in a more cautious and balanced manner.
Reviewer 3 Report
Comments and Suggestions for Authors
The manuscript titled *Psychometric Evidence of the Pap Smear Test and Cervical Cancer Beliefs Scale (CPC-28) in Aymara Women Sample from Chile* evaluates the validity and reliability of the CPC-28 scale among Aymara women in Chile, a population with low cervical cancer screening rates. The study confirms the six-factor structure of the CPC-28, demonstrating good model fit and reliability. However, the scale did not predict Pap test adherence, highlighting cultural and structural barriers. The research contributes valuable psychometric evidence for underrepresented indigenous populations and underscores the need for culturally tailored interventions. The strengths are: the study addresses a critical gap by validating the CPC-28 in an indigenous population, offering insights into cultural influences on health beliefs; robust methodological design, including stratified probabilistic sampling and confirmatory factor analysis (CFA), strengthens the findings; the discussion effectively contextualizes results within broader literature on health disparities and the Health Belief Model (HBM). The weaknesses are: predictive validity: the CPC-28’s inability to predict Pap test adherence raises questions about its utility in this population. The authors should explore alternative explanations (e.g., cultural factors not captured by HBM) or suggest modifications to the scale; cross-sectional design: causality cannot be inferred; longitudinal or mixed-methods approaches could better elucidate barriers and beliefs; missing controls: the analysis does not control for key confounders like healthcare access or education level, which may influence both beliefs and screening behavior.
In addition, I would like to mention some specific details: Table 3: Factor loadings for Item 26 (susceptibility dimension) are notably low (0.355). Consider revising or removing this item to improve reliability; Page 10, Table 5: The small effect size (d = 0.23) for "Need to undergo a Pap smear" suggests minimal practical significance, yet the trend is highlighted. Clarify the clinical relevance of this finding.
I can recommend to expand the discussion on cultural adaptations needed for the CPC-28 in indigenous contexts; to address low reliability in the susceptibility dimension, possibly via qualitative follow-up; to consider controlling for socioeconomic variables in future analyses.
A valuable contribution with minor revisions needed to enhance interpretability and impact. The study advances understanding of cervical cancer screening disparities in indigenous communities.
Author Response
The reviewer makes excellent points, all of them already contained across the other 3 reviewers, which have pointed out limitations and made comments that have improved this manuscript. Thus, and in response to the reviewer’s comments, all focused in the discussion section, we have revised the discussion section to incorporate a more critical reflection on the study’s methodological and conceptual limitations. In this updated version, we address concerns related to predictive validity, cross-cultural applicability of the CPC-28, issues of generalizability, and potential sources of bias. While these points are treated in a summarized fashion to maintain the flow of the discussion, we have taken care to ensure that the limitations are acknowledged and that the conclusions are now presented in a more cautious and balanced manner. To some of the other more specific comments, there are changes in the manuscript result section (based on other reviewer’s suggestions) the derived in new text which replaced some of the mistakes pointed out.
We kindly invite the reviewer to also consider the comments and our responses to the other review, as many of your suggestions were similarly addressed there—often in more detail or directly.
Reviewer 4 Report
Comments and Suggestions for Authors
Dear Authors,
I consider this kind of study important to understand the cultural and psychological factors influencing health behaviours. Evaluating tools like the CPC-28 can help researchers to better understand women's beliefs and attitudes towards cervical cancer and Pap tests. By doing so, tailored strategies can be developed to improve screening participation and address existing health inequities among vulnerable and underserved groups.
Nevertheless, while the article offers relevant findings, it would benefit from several improvements in structure and clarity, particularly in the introduction, to enhance coherence and facilitate a better understanding of the study’s context and rationale.
- The introduction lacks a clear and progressive structure. It begins directly with the main findings and study objectives, without first adequately contextualizing the problem under investigation. For greater clarity and impact, it is recommended to reorganize the introduction following a more conventional and logical structure: start by presenting the global context of cervical cancer, followed by national and local epidemiological data (Chile, Arica and Parinacota region), emphasizing the relevance of Pap smear screening and the specific challenges faced by the indigenous population. Additionally, the introduction shifts abruptly between global statistics, regional data, and methodological foundations, which hinders readability. Including transitional statements would help improve the text’s coherence and flow.
I suggest restructuring the introduction in a progressive manner, starting with the global relevance of the topic, moving through the national and regional context, and then clearly identifying the existing gap, namely the lack of psychometric validation of the CPC-28, concluding with a clear presentation of the study objectives.
- At Methodology section/Discussion
Evaluating the psychometric properties of a tool like CPC-28 with a sample of 299 participants aged 25 - 65 may present certain limitations. Women between 25-64 may have different beliefs, experiences, or barriers related to cervical cancer screening, which are not captured by this specific sample. This Must be discussed.
- Table 1, presenting detailed demographic data of participants, is placed in the methodology section. Conventionally, such tables should be included in the Results section, as they report collected data rather than methodological processes.
- The conclusion is written more as a discussion rather than a concise summary of the study's main findings and implications. You should briefly reinforce the key results, clarify the relevance of those results, and point to future research or applications, without extensively comparing with other studies.
The study appropriately adheres to ethical guidelines, including obtaining informed consent and approval from an ethics committee. The inclusion of intercultural facilitators demonstrates careful attention to cultural sensitivity and participant accessibility. Additionally, employing both digital and paper-based data collection methods ensures broader participation, and preliminary training sessions for facilitators reflect thorough methodological preparation. The statistical analyses conducted are well executed and appropriate for the study’s objectives. However, the CPC-28 instrument's limited predictive validity concerning actual screening behaviors suggests room for further methodological refinement.
Kind Regards.
Author Response
Introduction
The journal requests a plain language summary, which included in our introduction as its first paragraph. This was a mistake of edition that will be rectified. Furthermore, after removing citations, we tested the readability of the introduction using chat GPT. The results were as follows:
Flesch Reading Ease: 31.8 (Difficult to read; best understood by university graduates), Flesch-Kincaid Grade Level: 15.1 (College level), Gunning Fog Index: 17.4 (Requires 17+ years of formal education), SMOG Index: 14.6 (Approx. college level), Automated Readability Index (ARI): 13.8. This was interpreted and deemed as suitable for academic reading. However, and while keeping in mind and integrating other reviewer’s comments, we made special efforts in improving even more the readability of the introduction.
Methods
We kindly disagree with the reviewer regarding their comment on Table 1 being part of the results. Participants demographic are not part of the study goal.
When the reviewer states “The conclusion is written more as a discussion rather than a concise summary of the study's main findings and implications,” we are somewhat puzzled. Indeed, we wrote a discussion section, not a conclusion. In a discussion, we discuss at length the implications of the findings, and therefore, we believe we cannot satisfy the reviewer’s comment “concise summary of the study's main findings and implications,” because it contradicts several of the comments made by the other reviewers.
Discussion
We indeed agree with the reviewer on their point regarding the CPC-28 predicting validity. Along with the recommendations to other reviewers, we improve the implications in this regard. Finally, and to most of the comments made related to the discussion/conclusion section of the manuscript, which were very similar to several comments made by other reviewers, we have revised the discussion section to incorporate a more critical reflection on the study’s methodological and conceptual limitations. In this updated version, we address concerns related to predictive validity, cross-cultural applicability of the CPC-28, issues of generalizability, and potential sources of bias. While these points are treated in a summarized fashion to maintain the flow of the discussion, we have taken care to ensure that the limitations are acknowledged and that the conclusions are now presented in a more cautious and balanced manner.
We kindly invite the reviewer to also consider the comments and our responses to the other review, as many of your suggestions were similarly addressed there—often in more detail or directly.